# MOSAIC: Unlocking Over 30× Context Length for Diffusion LLMs Inference via Global Memory Planning and Dynamic Peak Taming

**Liang Zheng** [1]  **Bowen Shi** [1]  **Yitao Hu** [1]  **Jiawei Zhang** [1]  **Ruofan Li** [1]  **Guotao Yang** [1]  **Zhixin Zhao** [1]
**Zhengchao Wang** [1]  **Sheng Chen** [1]  **Wenxin Li** [1]  **Dezhi Ran** [2]  **Tao Xie** [2]  **Keqiu Li** [1]

## Abstract

Diffusion-based large language models (dLLMs) have emerged as a promising alternative to autoregressive models, leveraging simultaneous denoising to enable global planning and iterative refinement. These properties make dLLMs attractive for long-context generation. However, deploying dLLMs faces a prohibitive memory barrier, as existing inference systems are inefficient for the diffusion paradigm. We observe that current inference systems are misaligned with dLLMs. Unlike autoregressive models, whose memory footprint is dominated by the KV-Cache, dLLMs are bottlenecked by transient activations rematerialized per step. Moreover, generic memory reuse mechanisms lack the global visibility to handle dynamic memory peaks of dLLMs, which alternate between logits and feed-forward networks. To address these challenges, we present MOSAIC, a memory-efficient inference system that shifts dLLM execution from local, static management to a global, dynamic paradigm. MOSAIC integrates (i) a mask-only logits kernel eliminating redundant activation materialization, (ii) a lazy chunking optimizer using online heuristics to tame dynamic memory peaks, and (iii) a global memory manager leveraging virtual addressing to mitigate memory fragmentation. Evaluations show that MOSAIC reduces the memory peak-to-average ratio by 2.71× on average and increases the maximum inference sequence length on identical hardware by 15.30–32.34×. Crucially, MOSAIC is training-free and preserves exact model outputs, while reducing end-to-end latency by 2.5%–55.4%. Our code is publicly available at https://github.com/flashserve/Mosaic.

[1]Tianjin University, China [2]Peking University, China. Correspondence to: Yitao Hu <yitao@tju.edu.cn>.

*Proceedings of the 43rd International Conference on Machine Learning*, Seoul, South Korea. PMLR 306, 2026. Copyright 2026 by the author(s).

## 1. Introduction

While autoregressive (AR) models have driven recent breakthroughs in generative AI (Zhao et al., 2023; Minaee et al., 2024), diffusion-based large language models (dLLMs) have emerged as a distinct paradigm. Unlike the sequential, left-to-right generation of AR models, dLLMs progressively denoise the entire sequence simultaneously (Nie et al., 2025; Zhu et al., 2025; Ye et al., 2025). This capability facilitates global planning and iterative refinement, making dLLMs a promising paradigm for maintaining long-range consistency (Sahoo et al., 2024; Yu et al., 2025).

A key trend in dLLM is the shift toward long-context generation (Liu et al., 2025a; He et al., 2025). This capability unlocks advanced applications such as repository-level code generation (Xie et al., 2025) and massive-context information filling. Despite this potential, deploying long-context dLLMs remains constrained by a prohibitive memory capacity barrier stemming from system inefficiencies.

Specifically, existing memory optimizations for LLMs prioritize KV-Cache (Kwon et al., 2023; Prabhu et al., 2025), the primary bottleneck in autoregressive generation; however, these techniques do not transfer to dLLMs. Our profiling reveals a fundamental shift: dLLMs are bottlenecked by *transient activations* recomputed at every step. While general-purpose reuse mechanisms address activation memory to some extent (Pisarchyk & Lee, 2020; Ansel et al., 2024), they lack the global visibility to adapt to dLLMs' *dynamic peaks*, which toggle between logits and feed-forward networks (FFNs). Consequently, applying existing methods results in significant memory fragmentation and redundancy, necessitating a system tailored to these distinct patterns.

To materialize this shift, we propose MOSAIC, a memory-efficient inference system tailored for long-context dLLMs. MOSAIC shifts from local, static management to a global, dynamic paradigm. We introduce a mask-only logits kernel that eliminates redundancy by computing logits solely for masked tokens. To handle dynamic bottlenecks, a lazy chunking optimizer employs an online search to split memory-intensive operators only when necessary, minimizing latency overhead. Underpinning these components is a

global memory manager utilizing a graph registrar to capture the full computation lifecycle, enabling a unified virtual memory mapping that eliminates external fragmentation.

Our contributions are summarized as follows:

- We provide the first comprehensive characterization of dLLM memory usage, identifying the bottleneck shift to transient activations and the phenomenon of dynamic memory peaks.

- We propose MOSAIC, an inference system that integrates mask-only computation, a lazy chunking strategy, and a global memory manager utilizing a graph registrar to capture the full computation lifecycle for efficient long-context dLLM inference.

- Extensive evaluations demonstrate that MOSAIC achieves an average 2.71× reduction in memory peak-to-average ratio and extends the maximum supportable sequence length by 15.30–32.34×. Crucially, MOSAIC is training-free and preserves exact model outputs, while simultaneously reducing end-to-end latency by 2.5%–55.4%.

**Conflict of Interest Disclosure.** The authors declare that they have no financial conflicts of interest to disclose.

## 2. Preliminaries

This section reviews LLM inference paradigms and memory management techniques, focusing on the fundamental differences between autoregressive (AR) and diffusion-based LLMs (dLLMs) and their distinct memory characteristics.

### 2.1. From Autoregression to Diffusion

LLM inference has traditionally followed the autoregressive (AR) paradigm (Yang et al., 2025; Grattafiori et al., 2024). Recently, diffusion-based LLMs (dLLMs), such as LLaDA (Nie et al., 2025) and Dream (Ye et al., 2025), have emerged with fundamentally different inference workflows. Although both paradigms typically adopt the Transformer architecture (Vaswani et al., 2017), they exhibit distinct computation patterns and memory behaviors.

**Autoregression.** AR inference generates tokens sequentially. At each iteration, the model produces a single token, appends it to the sequence, and feeds the extended sequence into the next step. Due to causal dependencies, the effective sequence length grows monotonically until an EOS token or a length limit is reached.

**Diffusion.** In contrast, dLLMs perform non-autoregressive, iterative refinement, generating all output tokens jointly. Given a prompt, target length $L$, and inference steps $N$,

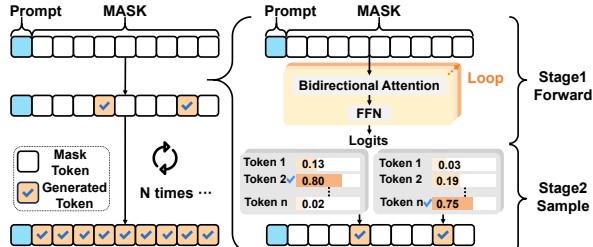

*Figure 1.* dLLM inference pipeline.

the process starts from a fully masked sequence of length $L$. Over $N$ steps, masks are progressively replaced with sampled tokens.

As shown in Figure 1, each step consists of two stages: (1) a bidirectional forward pass that produces logits for all masked positions, and (2) a sampling stage that selects and fills a subset of masks. Across iterations, the sequence transitions from fully masked to fully specified.

**Memory Implications.** Both paradigms produce large intermediate tensors, including activations and attention-related states, whose memory footprint often exceeds that of model parameters. Efficient memory management is therefore critical for scalable inference.

### 2.2. LLM Memory Management

LLM inference memory consumption is dominated by two categories: (1) transient activations within a forward pass, and (2) persistent caches such as the KV-Cache in AR decoding. We briefly summarize representative management techniques as follows.

**Transient Activations via Memory Reuse.** Activations are short-lived and only required within a single forward pass. Graph-based memory planning models the computation as a directed acyclic graph (DAG), enabling liveness analysis to identify each tensor's creation and last use. Tensors with non-overlapping lifetimes can safely share memory buffers, reducing peak memory usage. This technique underpins memory optimization in modern inference engines (Ansel et al., 2024; Pisarchyk & Lee, 2020; Lamprou et al., 2023).

**KV-Cache via Paging.** In AR inference, causal attention allows past keys and values to be reused across iterations, forming the KV-Cache (Li et al., 2024). Inference consists of a prefill phase that initializes the cache from the prompt and a decode phase that incrementally extends it. Since KV-Cache size grows linearly with sequence length, paging-based methods such as PagedAttention (Kwon et al., 2023) store KV tensors in fixed-size blocks, reducing fragmentation and enabling long-context generation.

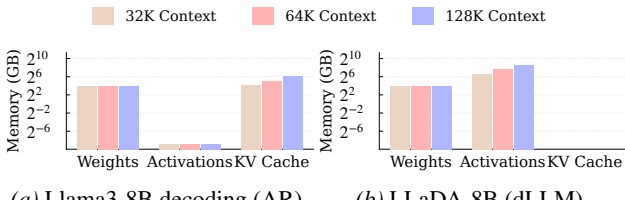

*(a)* Llama3-8B decoding (AR).  *(b)* LLaDA-8B (dLLM).

*Figure 2.* Memory breakdown of AR models and dLLMs.

## 3. Observations

As discussed in §2.1, efficient memory management is critical for long-context inference. However, existing techniques (§2.2), largely designed for autoregressive (AR) models, do not translate to diffusion-based LLMs (dLLMs). Through empirical analysis, we identify four observations that reveal fundamental mismatches between conventional memory management and dLLM inference.

**Observation #1: The memory bottleneck shifts from KV-Cache to transient activations.** We first compare the memory breakdown of AR and diffusion inference as context length increases. In AR decoding, transient activations are computed only for the newly generated token at each step, while the KV-Cache grows monotonically and dominates memory usage (Figure 2a).

In contrast, dLLMs employ bidirectional attention, causing key and value tensors to change across steps. This prevents caching and requires recomputing activations for all tokens at every step. As a result, transient activations dominate memory consumption in dLLMs (Figure 2b). This fundamental shift indicates that dLLM inference requires a *transient-activation–centric* memory design rather than the KV-Cache–centric paradigm used in AR systems.

**Observation #2: Logits computation wastes memory on unmasked tokens.** Given dominance of transient activations, we observe that existing dLLM inference systems still compute logits for all tokens at each step, though only masked tokens are sampled (Nie et al., 2025; Ye et al., 2025; Ma et al., 2025). This eager computation introduces substantial unnecessary memory overhead from unmasked logits.

Analogous to how KV-Cache avoids redundant recomputation in AR models, a natural optimization is *mask-only logits*, which restricts logits computation to masked tokens. However, this is non-trivial in practice, as logits computation requires memory-contiguous inputs (Gale et al., 2020), while masked tokens are typically scattered.

**Observation #3: dLLM inference exhibits dynamic memory peaks.** Even with mask-only logits, inference remains constrained by two transient memory peaks: FFN activations and logits. As shown in Figure 3, the dominant peak shifts dynamically with the mask ratio $r_m$ (the fraction of

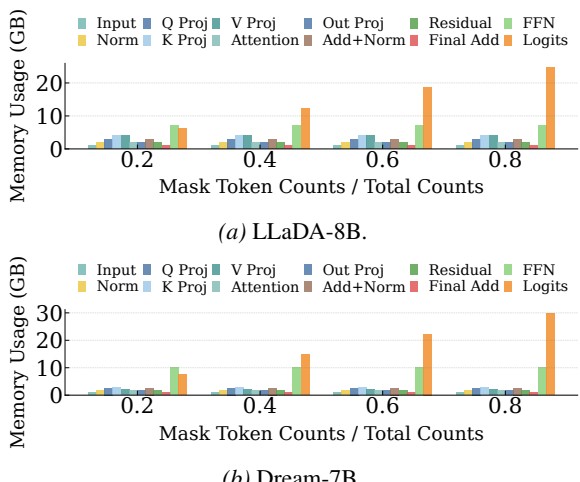

*(a)* LLaDA-8B.

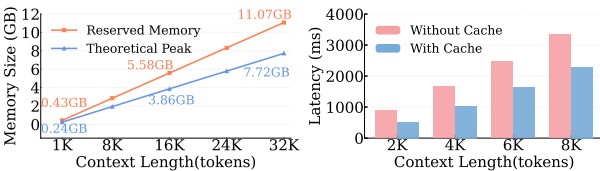

*(b)* Dream-7B.

*Figure 3.* Dynamic memory bottleneck shift across varying mask ratio $r_m$ at 128k context length.

*Figure 4.* Reserved memory vs. theoretical peak.  *Figure 5.* Latency w/ and w/o segment caching.

masked tokens). At low $r_m$, FFN activations dominate, whereas at high $r_m$, logits dominate. This behavior is consistent across models such as LLaDA-8B (Nie et al., 2025) and Dream-7B (Ye et al., 2025).

This dynamic arises from two factors: (1) large feature dimensions, where both FFN intermediate sizes and logits vocabulary sizes significantly exceed the hidden dimension; and (2) computation asymmetry, where FFNs operate on full sequence while logits are computed only for masked tokens. As $r_m$ varies, the relative contribution of these components shifts, forcing systems to provision memory for worst-case peak and thereby limiting achievable context length.

**Observation #4: Myopic memory planning causes severe fragmentation.** Unlike specialized AR inference engines, dLLM frameworks typically rely on `torch.compile` and a generic caching allocator (Paszke et al., 2019). We find that this combination incurs substantial memory overhead. As shown in Figure 4, at a 32k context length, reserved memory exceeds the theoretical peak by 43.19%.

The root cause is myopic memory planning induced by graph partitioning. Due to greedy grouping and fragile dynamic graph capture (Ansel et al., 2024; Liang et al., 2025), `torch.compile` frequently partitions execution into isolated subgraphs (Figure 6(a)). Each subgraph independently requests contiguous memory, preventing global reuse across

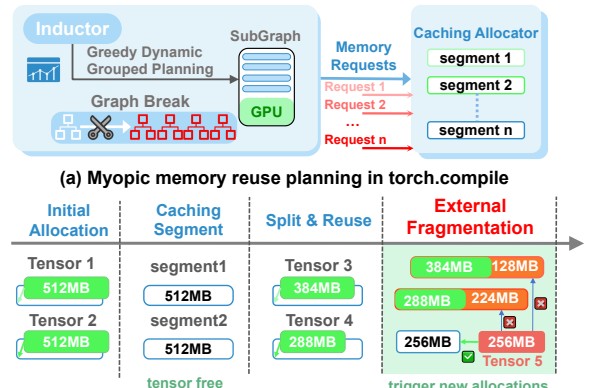

**(a) Myopic memory reuse planning in torch.compile**

**(b) Fragmentation issue due to PyTorch's segment mechanism**

*Figure 6.* Illustration of (a) myopic memory reuse planning and (b) the resulting memory fragmentation within the caching allocator.

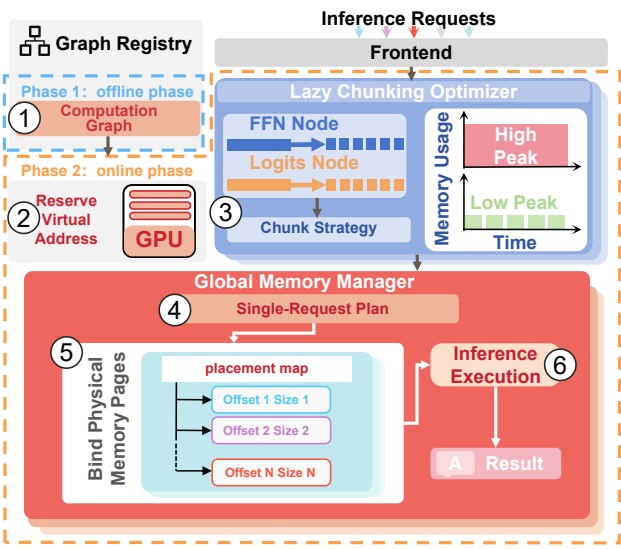

*Figure 7.* MOSAIC overview.

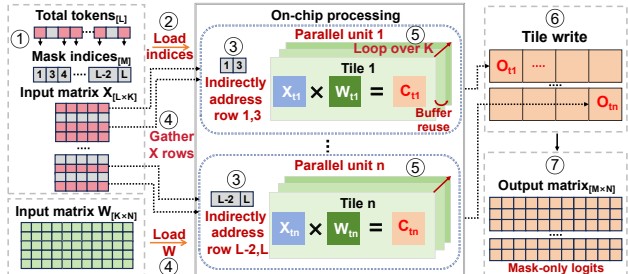

*Figure 8.* MOSAIC's mask-only logits kernel.

four tightly integrated components: (1) a *mask-only logits kernel* that computes logits only for masked tokens without auxiliary buffering; (2) a *graph registrar* that explicitly defines a parameterized computation graph to enable whole-graph visibility; (3) a *lazy chunking optimizer* that adaptively chunks memory-intensive operators (FFN and logits) to reduce peak memory with minimal latency overhead; and (4) a *global memory manager* that maps tensors into a virtually contiguous workspace to eliminate fragmentation.

As shown in Figure 7, MOSAIC operates in two phases. In the offline phase, the graph registrar constructs a parameterized graph template using symbolic primitives once per model ①. In the online phase, the global memory manager reserves a contiguous virtual address space ②. Upon receiving an inference request, the lazy chunking optimizer selects a chunking configuration via a bottleneck-driven search ③. The memory manager then derives a global tensor placement map ④, binds physical pages according to the planned peak memory ⑤, and executes inference ⑥.

We next describe the four components in detail.

### 4.2. Mask-only Logits Kernel

To realize mask-only logits (Observation #2), MOSAIC must address a key challenge: logits computation is a GEMM operation that requires memory-contiguous inputs, whereas masked tokens are typically scattered. A naive gather-then-compute approach necessitates a temporary contiguous buffer for the masked hidden states, introducing a memory overhead of $O(N_{\text{mask}} \times D)$, where $N_{\text{mask}}$ is the number of masked tokens and $D$ is the hidden dimension, creating significant redundancy.

MOSAIC addresses this with a fused gather–GEMM kernel. As illustrated in Figure 8, the kernel first extracts mask indices ① and loads them into on-chip memory ②. Using indirect addressing ③, it gathers scattered input rows and fetches weight tiles ④ directly into on-chip memory for iterative matrix multiplication ⑤. The accumulated results are written back directly to the output ⑥, forming the final logits ⑦ without intermediate buffering. Furthermore, the kernel reduces end-to-end latency by skipping non-masked

the full execution.

This fragmentation propagates to the physical allocator. Dynamic allocation (*e.g.,* via `cudaMalloc`) incurs high runtime overhead, increasing latency by 46%–72% (Figure 5). Although the allocator caches segments to reduce allocation cost, it preferentially reuses cached segments rather than allocating exact-sized buffers when handling sequential memory requests. This leaves behind non-contiguous fragments (Figure 6(b)) that cannot satisfy large tensor requests, forcing extra allocations and inflating reserved memory.

## 4. Design of MOSAIC

### 4.1. Overview

We propose MOSAIC, a system that enables long-context inference for diffusion-based LLMs. MOSAIC consists of

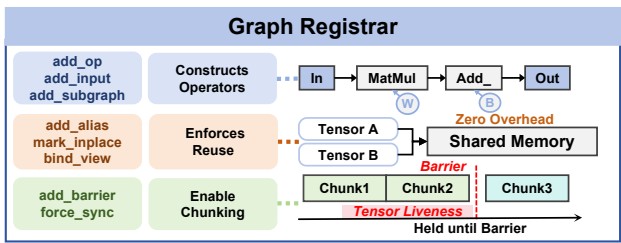

Figure 9. MOSAIC's graph registrar.

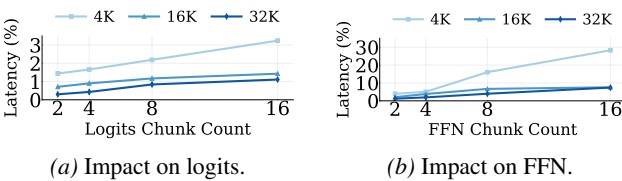

*(a)* Impact on logits.  *(b)* Impact on FFN.

Figure 10. Impact of context length and chunk count on the latency of logits and FFN.

tokens, with details in §5.2.

### 4.3. Graph Registrar

Supporting lazy chunking and global memory planning requires a complete view of the computation graph. However, compiler-level dynamic capture is often fragile and prone to graph breaks (Ansel et al., 2024; Liang et al., 2025). MOSAIC therefore introduces a graph registrar that explicitly defines a parameterized graph template.

As shown in Figure 9, the registrar uses symbolic primitives (*e.g.,* add_op) to construct a template that captures model structure and memory dependencies, while leaving input-dependent dimensions (*e.g.,* sequence length) symbolic and instantiated at runtime. This guarantees an unbroken, whole-graph view. The registrar also provides constraint primitives, such as add_alias, to enforce memory aliasing between operators, and chunking-related primitives to define chunked execution. Control primitives (*e.g.,* add_barrier) extend tensor lifetimes across chunk loops, preventing premature memory reclamation before all chunks complete.

### 4.4. Lazy Chunking Optimizer

As shown in §3, the memory bottleneck in dLLM inference shifts dynamically between logits and FFN depending on the mask ratio $r_m$. While chunking these operators along the sequence dimension can reduce peak memory, naive chunking suffers from two issues: (1) Latency overhead, as decomposing a single kernel into multiple launches adds execution cost. (2) Misaligned configuration, where static policies might chunk the wrong component as the bottleneck moves (*e.g.,* chunking FFN when logits is the peak).

To address these two challenges, MOSAIC proceeds in three stages. First, it analyzes the latency impact to identify the opportunity to chunk at long context. Second, guided by this, it introduces a lazy chunking strategy to minimize latency overhead. Third, to materialize this strategy under dynamic bottlenecks, we devise an online bottleneck-driven search to find the minimal sufficient chunk configuration.

**Opportunity to chunk at long context.** Figure 10 reveals that latency increase incurred by chunking diminishes at long contexts. For instance, at 32K context with 4 chunks, the overhead is less than 0.5% for logits and 2% for FFNs. This is because at long contexts, the execution time is dominated by massive matrix computations, rendering the fixed kernel launch overhead negligible by comparison.

**Lazy chunking strategy.** Based on this observation, we introduce a lazy chunking strategy. To minimize overhead, we only activate chunking when physical memory is insufficient. Since memory shortages primarily occur at long context, chunking is low-overhead.

When activated, this strategy partitions the logits and FFN computations into smaller segments to reduce peak memory. We denote the number of chunks for these operations as $K_{\text{logits}}$ and $K_{\text{FFN}}$, respectively. To ensure efficiency, we must identify the minimal configuration that exactly supports the target context. However, determining this configuration is non-trivial because the memory bottleneck shifts dynamically with the mask ratio $r_m$. Consequently, we employ an online bottleneck-driven search to solve this dilemma.

**Online bottleneck-driven search.** To instantiate the lazy chunking strategy, we need to find a feasible chunk setting. A naive solution is to brute-force search all combinations. However, this incurs prohibitive overhead. Crucially, we observe that due to memory reuse, the footprint is dictated by the peak and reducing non-peak components yields zero gain. Leveraging this insight, we prune the search space via a bottleneck-driven heuristic. To ensure consistency with runtime execution, we simulate exact buffer sizes by invoking the global memory planner on the registrar's graph.

As detailed in Algorithm 1, the search takes the number of masks and total tokens as inputs. The search iteratively identifies the current peak component, increases its chunk count, and re-evaluates memory usage using the global memory planner. The process terminates once the peak fits within available memory or no further chunking is possible. As validated in §5.3, MOSAIC with this heuristic matches the maximum supportable context length of MOSAIC with brute-force search, while the search latency of this heuristic is only 0.28%–0.32% of the brute-force search. The derived $K_{\text{logits}}$ and $K_{\text{FFN}}$ are applied to runtime execution.

---

**Algorithm 1:** Online Bottleneck-Driven Search

**Input:** Sequence length $T$, mask number $N_{\text{mask}}$, memory capacity $M_{\text{cap}}$

**Output:** Chunk configuration $(K_{\text{logits}}, K_{\text{FFN}})$ or failure

1  Initialize $K_{\text{logits}} \leftarrow 1, K_{\text{FFN}} \leftarrow 1$;
2  $P \leftarrow \text{SIMULATEPEAK}(T, N_{\text{mask}}, K_{\text{logits}}, K_{\text{FFN}})$;
3  **while** $P > M_{cap}$ **do**
       // Identify component causing the peak
4      Identify component $C_{\text{peak}}$ causing the current peak $P$;
       // Check if the peak component supports chunking
5      **if** $C_{peak} \notin \{\text{LOGITS}, \text{FFN}\}$ **then**
           // Insufficient memory capacity
6          **return** failure;
       // Adjust configuration based on bottleneck
7      **if** $C_{peak} = \text{LOGITS}$ **then**
8          $K_{\text{logits}} \leftarrow K_{\text{logits}} + 1$;
9      **else if** $C_{peak} = \text{FFN}$ **then**
10         $K_{\text{FFN}} \leftarrow K_{\text{FFN}} + 1$;
11     $P \leftarrow \text{SIMULATEPEAK}(T, N_{\text{mask}}, K_{\text{logits}}, K_{\text{FFN}})$;
   // Return valid configuration
12 **return** $(K_{\text{logits}}, K_{\text{FFN}})$;

---

### 4.5. Global Memory Manager

To eliminate the memory fragmentation described in §3, we identify the root cause of this fragmentation as issuing multiple memory requests. This mechanism implicitly delegates the mapping of logical requests to physical segments to the underlying allocator. As the allocator lacks a global view, this delegation inevitably leads to fragmentation.

To avoid fragmentation, we issue a one-shot global allocation to explicitly bypass the dynamic segment mapping of the underlying allocator. To enable this one-shot allocation, we treat the workspace as a unified address space and pre-calculate a tensor placement map specifying the offset and size for every tensor to direct the runtime execution.

MOSAIC implements this via two core components: (1) a single-request planner, which performs whole-graph analysis to derive a tensor placement map; and (2) a VMM-based allocator, which dynamically maps physical pages to the planned addresses.

**Single-request planner.** Constructing a unified placement map requires tracking tensor lifecycles. We achieve this by deriving the computation graph from the registrar's templates with lazy chunking. This visibility enables us to perform reuse planning, determining all tensors' offsets.

While finding the optimal offsets is an NP-hard problem solvable via integer linear programming (ILP) (Steiner et al., 2023), our evaluation (§5.3) indicates that ILP solvers incur prohibitive latency. Therefore, we adopt a first-fit heuristic, similar to existing works (Chen et al., 2018; Lin et al., 2020), which reduces planning time to 0.1%-4.3% of the ILP while

achieving the same supportable context length.

**VMM-based allocator.** Since the planner maps all tensors into a unified address space, the workspace buffer must be contiguous to support it. However, a naive static pre-allocation would lead to substantial memory waste when handling short sequences. To resolve this, MOSAIC leverages a virtual memory management (VMM) allocator.

Specifically, during initialization, we only reserve a large range of contiguous virtual addresses occupying no physical memory. At runtime, once the planned peak size is known, we dynamically bind physical pages to this virtual range using low-level page mapping APIs. This ensures that the execution sees a contiguous buffer while the physical consumption is strictly limited to the actual needs.

With the physical memory being ready, MOSAIC executes inference following the tensor placement map. However, standard PyTorch operators employ opaque memory management (*e.g.,* implicitly allocating internal buffers and lacking support for explicit output placement) that obstructs the strict execution of our placement map. To resolve this, we replace them with customized placement-aware operators to enforce absolute control over memory allocation. These operators not only write directly to pre-determined memory locations but also eliminate redundant internal buffering. For instance, it places Dream (Ye et al., 2025)'s logits shift and concatenation with a token-level kernel that writes partial results directly to their final positions, which reduces memory usage.

## 5. Evaluation

### 5.1. Experimental Setup

**Implementation.** We implemented MOSAIC based on vLLM 0.10.1 (Kwon et al., 2023) to use its robust serving capability. To enable dLLM inference, we integrated the official `PyTorch` model and bypassed vLLM's KV-Cache components, replacing them with dLLM-specific logic. MOSAIC supports variable-length inference to eliminate padding and implements iteration-level scheduling for efficient execution. MOSAIC comprises approximately 11k lines of code, consisting of 9.5k lines of Python, 1.3k lines of C++/CUDA, and 0.5k lines of Triton.

**Hardware Setup and Models.** We evaluated MOSAIC on an NVIDIA RTX 3090 with 24GB VRAM and an NVIDIA A100 with 40GB VRAM. We select three mainstream dLLMs: LLaDA-8B (Nie et al., 2025), Dream-7B (Ye et al., 2025), and LLaDA-MoE (Zhu et al., 2025). As MOSAIC targets low-level memory management, its performance is associated with computation graphs and tensor shapes, remaining agnostic to semantic content. Therefore, we evaluate workloads composed of varying prompt and total lengths

to cover diverse memory patterns.

**Baselines.** We compare MOSAIC against three baselines: (1) PYNATIVE: The unmodified official `PyTorch` code for dLLM inference. This measures the raw performance without any serving infrastructure; (2) MOSAIC-TORCH: The `PyTorch` model code ported onto the vLLM framework to leverage its robust serving infrastructure, but without memory optimizations; (3) MOSAIC-COMPILE: This baseline integrates `torch.compile` into the MOSAIC-TORCH.

**Key Metrics.** We evaluate MOSAIC and baselines on two metrics: (1) Per-step latency[1], the average time per step; (2) Maximum context length $L_{\max}$, the largest context size supportable before an out-of-memory error.

### 5.2. End-to-end Evaluation

For a comprehensive evaluation, we selected requests with varying ratios of prompt length over total length, denoted as $r_p$, for each model. Figure 11 illustrates the maximum supportable context length of MOSAIC and the baselines. As validated in Appendix A.1, our porting to vLLM incurs no performance penalty; separately, we note that MOSAIC-COMPILE extends $L_{\max}$ but suffers from compilation latency. Against this backdrop, MOSAIC consistently outperforms all alternatives in $L_{\max}$. On average across the three evaluated models and two hardware configurations, its $L_{\max}$ is 32.34, 25.41, and 15.30 times that of PYNATIVE, MOSAIC-TORCH, and MOSAIC-COMPILE, respectively. To understand the $L_{\max}$ difference, we conjecture that MOSAIC's advantage arises from two key factors as follows.

First, existing systems fail to handle memory peaks, thereby bounding the context capacity. In contrast, MOSAIC employs lazy chunking to mitigate this issue by chunking peak activations when memory is insufficient for longer contexts. As shown in §A.2, we utilize the memory's peak-to-average ratio to quantify the fluctuation of activation memory usage. MOSAIC exhibits an average 2.71× reduction in ratio when chunking is enabled; notably, the ratio is lower even before chunking activates because the mask-only kernel reduces logits memory. This effectively resolves spikes, which directly translates to a higher $L_{\max}$.

Second, existing systems rely on myopic memory planning, leading to external fragmentation and inflated reserved memory compared to the theoretical peak, evidenced by a 21.78%–78.33% inflation rate in Figure 12. In contrast, MOSAIC employs a global manager to eliminate external fragmentation and optimize operators for placement-aware execution (*e.g.,* replacing Dream's unique logits shift with a memory-saving, token-level alternative), collectively en-

abling a larger $L_{\max}$.

Besides, MOSAIC incurs no compromise on latency. It achieves a 2.5%–24.9% reduction compared to MOSAIC-TORCH and MOSAIC-COMPILE, attributed to the mask-only logits kernel eliminating redundant computations and the graph registration system bypassing the dynamic graph capture overhead inherent in `torch.compile`. Furthermore, MOSAIC reduces latency by 5.9%–55.4% compared to PYNATIVE; this significant gain stems from the aforementioned optimizations combined with the inherent engineering efficiencies of the vLLM framework.

### 5.3. Ablation Study

We ablate MOSAIC 's components to quantify their contributions to maximum context length $L_{\max}$ and latency. Due to space limits, we report results for the prompt-to-total ratio $r_p$=0.5; other $r_p$ show similar trends.

**Breakdown of $L_{\max}$ improvements.** To quantify each component's contribution to $L_{\max}$, we performed a cumulative breakdown of the performance gains. For each baseline, we averaged improvements across models and hardware. The reported ranges span the results across the three baselines by incrementally integrating the global memory manager, mask-only kernel, and lazy chunking optimizer.

As shown in Figure 13, the global manager increases $L_{\max}$ to 1.41×–3.00× that of the baselines through global memory reuse coupled with optimized placement-aware operators. Incorporating the mask-only kernel further elevates the capacity to 3.68×–8.04× relative to the baselines by eliminating redundant logits memory for unmasked tokens, while lazy chunking ultimately boosts $L_{\max}$ to 15.30×–32.34× over the baselines by mitigating memory spikes.

**Impact of chunking strategy.** MOSAIC employs a lazy chunking strategy to mitigate the latency overhead induced by chunking operations while maintaining the same $L_{\max}$. To evaluate this, we compare MOSAIC against MOSAIC-FC, a baseline that statically applies the configuration supporting $L_{\max}$ across all context lengths.

Our evaluation confirms that both strategies achieve identical $L_{\max}$. As shown in Figure 14, MOSAIC achieves a 5.5%–21.5% latency reduction within the 32k context range compared to MOSAIC-FC. This reduction is particularly pronounced at shorter context lengths, where the relative overhead of chunking is higher. By dynamically disabling chunking for short contexts, MOSAIC effectively eliminates this unnecessary overhead.

**Efficiency of bottleneck-driven search.** We compare MOSAIC 's bottleneck-driven search with MOSAIC-BF, which uses brute-force search. Both reach the same $L_{\max}$, while MOSAIC 's search latency is only 0.28%–0.32% of brute-

---

[1]Unlike autoregressive models, dLLMs operate via iterative denoising steps without distinct prefill and decode phases, rendering metrics like TTFT and TPOT inapplicable.

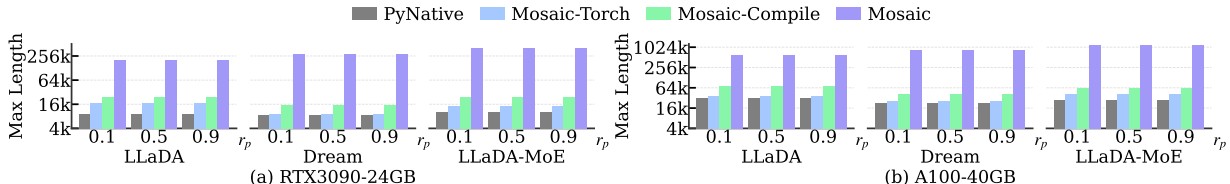

Figure 11. End-to-end performance evaluation comparing $L_{max}$ of MOSAIC and baselines across prompt-to-total ratios $r_p$.

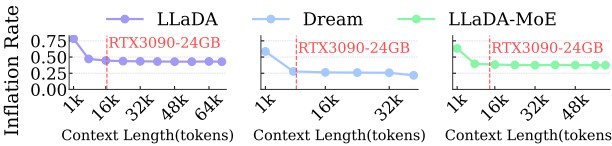

Figure 12. Inflation rates of baselines.

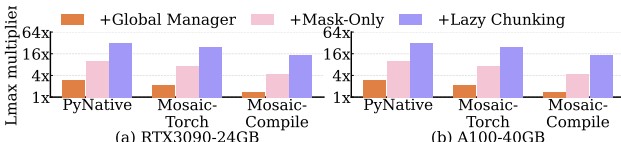

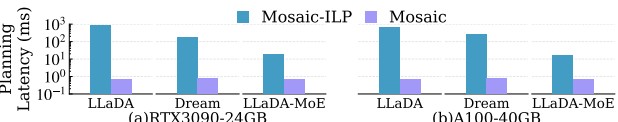

Figure 16. Comparison of planning latency.

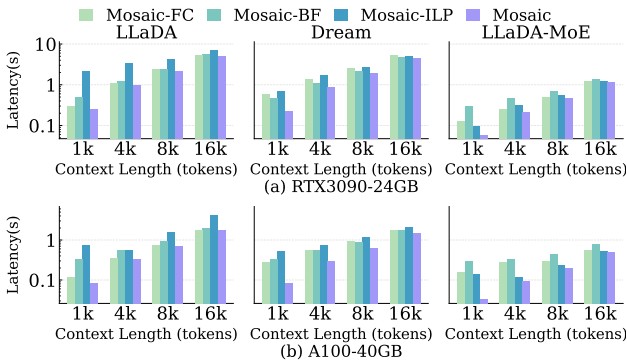

Figure 13. Impact of the global memory manager, mask-only kernel, and lazy chunking on $L_{max}$.

Figure 14. Impact of search, planning, and chunking strategies on end-to-end latency.

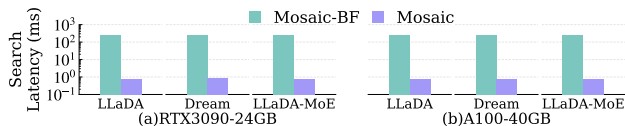

Figure 15. Comparison of search latency.

force (Figure 15). This reduces short-context latency by 22.3%–70.7% relative to MOSAIC-BF (Figure 14); the gap narrows at long contexts where computation dominates.

**Efficiency of first-fit planning.** MOSAIC employs a first-fit strategy to determine tensor offsets. We compare it against MOSAIC-ILP, a baseline utilizing integer linear programming (ILP) for theoretically optimal planning. Our evalua-

tion confirms that both achieve the same $L_{max}$, while first-fit reduces planning latency to 0.1%–4.3% of ILP (Figure 16). This translates to a 7.1%–36.7% reduction in overall latency versus MOSAIC-ILP (Figure 14).

### 5.4. Scalability and Compatibility

Having demonstrated the effectiveness of MOSAIC's core components, we further evaluate its robustness in broader, real-world deployment scenarios. Specifically, to verify MOSAIC's scalability across hardware configurations and high-concurrency workloads, as well as its compatibility with existing tailored inference stacks, we extend our evaluation to the following three advanced settings.

**Scalability to Multi-GPU Settings.** We extended MOSAIC to support multi-GPU inference environments. Evaluated on the LLaDA model across 2 GPUs, MOSAIC achieves a 12.30× and 14.92× context length extension under Tensor Parallelism (TP2) and Pipeline Parallelism (PP2), respectively, compared to the PYNATIVE baseline.

In distributed inference settings, the overall expansion ratio is naturally influenced by the necessary communication buffers. As the context scales to extreme lengths, the memory footprint of these fixed communication buffers (e.g., for inter-GPU tensor passing) grows linearly. While MOSAIC effectively eliminates the computation activation bottleneck, these unoptimizable communication buffers eventually consume a significant proportion of the GPU memory, inherently bounding the theoretical upper limit of the current context extension. Nevertheless, MOSAIC still delivers meaningful capacity gains, demonstrating its robustness under distributed environments.

**Compatibility with KV-Cache Frameworks.** While MO-SAIC achieves maximum context extensions of over 32× on standard lossless baselines where intermediate activations form the strict memory bottleneck, it remains highly

complementary to tailored KV-cache stacks. Unlike standard lossless execution, integrating KV caching into dLLMs typically requires lossy approximations. This structurally shifts the memory bottleneck from transient activations to persistent KV states, inherently reducing the optimization scope for activation memory management.

Nevertheless, by porting MOSAIC to specialized frameworks, namely Fast-dLLM (Wu et al., 2025), dInfer (Ma et al., 2025), and Elastic-Cache (Nguyen-Tri et al., 2025), we demonstrate its robust compatibility. Evaluated on an NVIDIA RTX 3090 with 24GB VRAM using the LLaDA-8B model, MOSAIC still extends the supported context lengths by 3.75×, 3.00×, and 2.27×, respectively. Furthermore, MOSAIC achieves these capacity extensions under identical hardware configurations without incurring any latency degradation.

**Performance on Batched Workloads.** To validate the practical utility of MOSAIC under real-world serving scenarios, we further evaluate its scalability with batched workloads. In high-concurrency environments, the memory footprint of intermediate activations scales linearly with the total number of processed tokens. Consequently, under a fixed hardware memory ceiling and a uniform sequence length across all requests, extending the maximum context length mathematically equates to expanding the batch capacity.

Our empirical evaluations confirm this theoretical equivalence. Specifically, when testing with identically sized requests, MOSAIC increases the maximum supported batch size by the exact same multipliers observed in the single-request context extensions, achieving an average increase of 32.34×, 25.41×, and 15.30× over PYNATIVE, MOSAIC-TORCH, and MOSAIC-COMPILE, respectively. Furthermore, within the executable range of these baselines, MOSAIC gracefully handles batched execution without incurring any latency penalty.

## 6. Related Work

**Diffusion Language Models and Optimization.** Text diffusion has evolved from continuous approximations (Li et al., 2022; Gong et al., 2022) to discrete token-space formulations (Austin et al., 2021; Han et al., 2023), culminating in Transformer-based dLLMs (Nie et al., 2025; Ye et al., 2025) that leverage bidirectional attention. While one line of work uses KV-Cache (Wu et al., 2025) to accelerate inference, it re-introduces the persistent KV bottleneck and may compromise generation quality (Wu et al., 2025; Liu et al., 2025b; Hu et al., 2025b). In contrast, we target the standard bidirectional dLLMs to ensure strict global consistency.

**Activation Memory Reuse.** Mainstream frameworks optimize memory via liveness analysis (Pisarchyk & Lee, 2020; Ansel et al., 2024). However, fragile dynamic capture often

triggers "graph breaks" (Ansel et al., 2024) that produce isolated sub-graphs, resulting in suboptimal reuse.

**Persistent State Memory Optimization.** Systems like vLLM (Kwon et al., 2023) and vAttention (Prabhu et al., 2025) optimize AR inference by managing the persistent KV-Cache via paging or virtual memory. Additionally, KV-Cache offloading alleviates the memory pressure (Hu et al., 2025a), while prefix-aware attention accelerates inference by aggregating KV states (Yi et al., 2025). However, these are ill-suited for bidirectional dLLMs.

## 7. Conclusion

In this paper, we identified the fundamental mismatch between conventional memory management and the unique requirements of dLLMs, specifically the shift from KV-Cache to transient activations and dynamic memory peaks. To bridge this gap, we proposed MOSAIC, a system that harmonizes mask-only computation, global graph planning, and adaptive lazy chunking. It effectively eliminates redundancy and fragmentation, enabling 15.30–32.34× longer context length and 2.5%–55.4% lower latency. Crucially, MOSAIC is training-free and preserves exact model outputs. Our work paves the way for scalable dLLM inference, making long-context applications more accessible and efficient.

## Acknowledgements

This work was supported by the National Natural Science Foundation of China under Grant 62572341.

## Impact Statement

This paper presents work whose goal is to advance the field of Machine Learning. There are many potential societal consequences of our work, none which we feel must be specifically highlighted here.

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

# A. Appendix

## A.1. Porting Validation and Latency Analysis

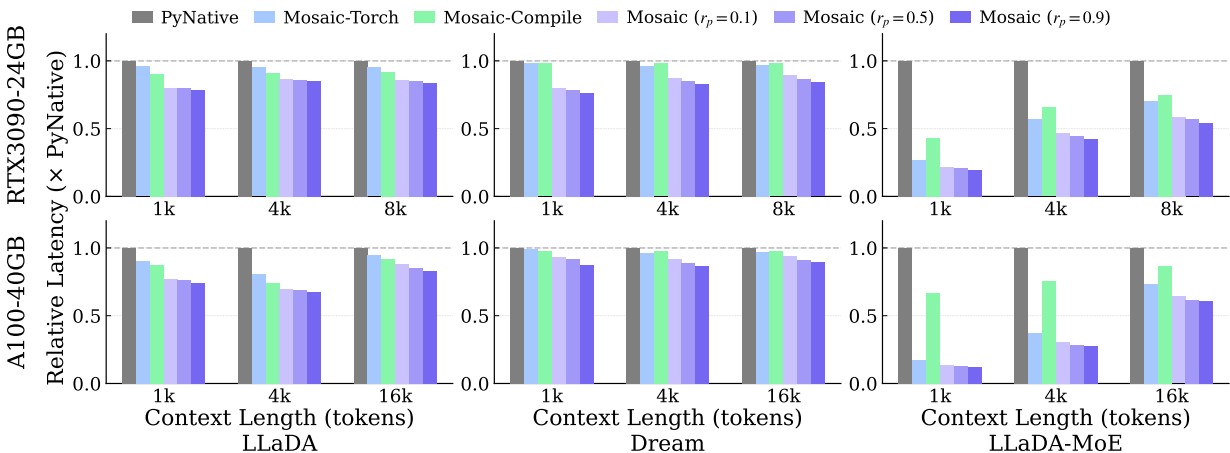

*Figure 17.* Latency comparison validating that the porting to vLLM incurs no performance penalty, while MOSAIC-COMPILE trades latency for increased context length.

**Porting validation.** As shown in Figure 17, MOSAIC-TORCH significantly outperforms PYNATIVE. Leveraging vLLM's efficient infrastructure such as operator optimizations, it not only reduces latency by 3.1%–41.2% but also increases the maximum context length $L_{max}$ by 7.7%–48.1%, as shown in Figure 11. These results confirm that our porting to vLLM incurs no performance penalty and yields comprehensive performance gains.

**Latency analysis.** When further enabling `torch.compile` on MOSAIC-TORCH, we observe a trade-off. Although MOSAIC-COMPILE extends $L_{max}$ by 45.5%–97.1%, it suffers from latency regression due to the overhead associated with dynamic graph capture and compilation, causing its latency to exceed that of MOSAIC-TORCH. In contrast, MOSAIC incurs no compromise on latency. It achieves a 2.5%–24.9% reduction compared to MOSAIC-TORCH and MOSAIC-COMPILE, and a 5.9%–55.4% reduction compared to PYNATIVE. This significant gain stems from the mask-only logits kernel eliminating redundant computations and the graph registration system bypassing the dynamic graph capture overhead inherent in `torch.compile`, combined with the inherent engineering efficiencies of the vLLM framework.

## A.2. Peak-to-Average Ratio Analysis

To further analyze the memory characteristics of dLLMs, we utilize the memory's peak-to-average ratio (PAR) to quantify the fluctuation of activation memory usage. A higher PAR indicates severe memory spikes that limit the maximum supportable context length $L_{max}$.

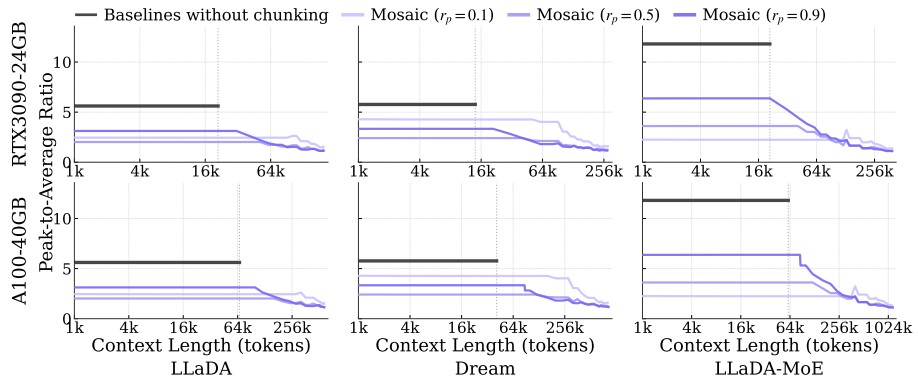

*Figure 18.* Comparison of activation memory peak-to-average ratio (PAR) across context lengths. Baselines without chunking refer to PYNATIVE, MOSAIC-TORCH, and MOSAIC-COMPILE.

As shown in Figure 18, MOSAIC exhibits a significantly lower ratio when chunking is enabled compared to baselines. Notably, the ratio is lower even before chunking activates because the mask-only kernel reduces logits memory. This reduction in PAR confirms that MOSAIC effectively smooths out memory spikes, allowing the system to support significantly longer contexts within the same memory budget.

