# OpenReview forum: "Mosaic: Unlocking Over 30$\times$ Context Length for Diffusion LLMs Inference via Global Memory Planning and Dynamic Peak Taming"
_ICML.cc/2026/Conference — ICML 2026 regular_

### Official Review · Reviewer_EE58 · 2026-03-12

**Soundness:** 3
**Presentation:** 2
**Significance:** 2
**Originality:** 2
**Overall Recommendation:** 4
**Confidence:** 3

**Summary:**

This paper proposes MOSAIC, a memory-efficient inference system for dLLMs (diffusion LLMs). This work features a mask-only logits kernel, lazy chunking for peak-memory control, and a global memory manager based on virtual addressing. The goal of this work is to support longer contexts for dLLM inference without changing model outputs, and the paper reports large gains in maximum supported context length together with modest or sometimes substantial latency improvements.

**Compliance With Llm Reviewing Policy:**

Affirmed.

**Final Justification:**

My main concerns regarding soundness have been resolved in the rebuttal. Although the main techniques used in the paper still feel incremental, I think the observations of the current methods' limitations and the practicality of the proposed system are valuable.

The authors promise to revise the paper to reflect the changes, and I believe the quality of this paper can be much improved with these changes. Therefore I increase my score to a positive one.

**Key Questions For Authors:**

1. Direct Framework Comparisons: How does MOSAIC compare in terms of $L_{max}$ and latency to dLLM-specific optimizations mentioned in the related work, such as dinfer?


2. Online Phase Latency: In real-world serving scenarios where the batch size is not 1, how does the system perform?


3. Kernel Generalization: Does the mask-only logits kernel require manual re-implementation for different model architectures (e.g., MoE vs. dense), or is it handled automatically by the graph registrar?

**Limitations:**

No, this paper does not discuss limitations. There is a small paragraph of impact statement, but it is very vague.
I suggest discussing which types of workloads that the system can or cannot perform well.

**Strengths And Weaknesses:**

**Strengths**

1. The paper addresses a practical systems problem. Memory-efficient inference for long-context diffusion LLMs is important, and the paper identifies a real mismatch between AR-oriented memory systems and dLLM inference.
2. The observations are interesting and there are significant engineering efforts involved.

**Weaknesses**

1. (Originality) The core ideas seem incremental. The major methods used to optimize the performance seem to be standard ideas in systems community, such as avoiding useless logits computation, chunking large operators, and global memory planning. The paper reads much more like an implementation package than a research advance.

2. (Soundness) The experiments are relatively limited in terms of the following aspects:
    - The comparison is largely limited to basic ports to vLLM, lacking comparisons against other dLLM-specific frameworks (e.g., dinfer https://arxiv.org/abs/2510.08666, Fast-dLLM https://arxiv.org/pdf/2505.22618, etc. ).
    - The evaluation primarily focuses on single-request scenarios with a batch size of 1, which may not fully reflect real-world deployment conditions. In high-concurrency or batch processing environments, the system's performance, especially in terms of latency and memory management, could be significantly different. The paper lacks detailed performance evaluations in scenarios with larger batch sizes or high-frequency requests, where memory and computation management optimizations such as lazy chunking and graph registrar could introduce additional overhead. A more comprehensive evaluation, including high-concurrency or batch-based workloads, is needed to better assess the system's scalability and performance in practical, real-world applications.

3. (Presentation) The abstract contains multiple paragraphs that should be consolidated into a single one.

---

> ### Author Rebuttal · Authors · 2026-03-31
>
> We thank the reviewer for the constructive feedback. Supporting long context is critical for AI agents, yet standard systems struggle with the unique memory challenges of dLLMs. We developed Mosaic to address these specific bottlenecks.  Guided by your feedback, we have comprehensively updated the manuscript as follows.
>
> **1. Originality and Research Advance (Weakness 1)**
>
> While operator chunking and memory planning are common, standard techniques fail for dLLMs. Our core advance is a tailored architectural co-design:
>
> *   **First Comprehensive Characterization:** We provide the first analysis showing dLLM bottlenecks shift from KV-cache to transient activations, with dynamic peaks toggling between logits and FFNs based on mask ratios.
> *   **Identifying General Method Limitations:** We identify that existing methods lack the global visibility to manage these shifting peaks, causing severe fragmentation.
> *   **Global Architectural Co-design:** We propose a global, dynamic framework that resolves the mismatch between conventional systems and dLLM-specific computation patterns.
>
> We will explicitly highlight these core advances and the necessity of our dLLM-specific architecture in the revision.
>
> **2. Comparisons with dLLM-Specific Frameworks (Weakness 2.1 & Key Question 1)**
>
> We categorized dLLM inference setups into two paradigms to provide a comprehensive evaluation against tailored systems:
>
> | Method Name                                    | Official PyTorch | JetEngine | Fast-dLLM | dInfer | Elastic-Cache |
> | :--------------------------------------------- | :--------------- | :-------- | :-------- | :----- | :------------ |
> | Context Length Extension (Mosaic vs. Baseline) | 32.34x           | 33.4x     | 3.75x     | 3.00x  | 2.27x         |
>
> *   **Achieving >32x context extensions on standard lossless baselines.** Standard implementations (official PyTorch, JetEngine [1]) lack KV caches, making intermediate activations their strict memory bottleneck (Sec. 3 in submission). By systematically eliminating unmanaged activation peaks and fragmentation via lazy chunking and global memory management, Mosaic prevents early Out-Of-Memory (OOM) errors. Evaluated on a 24GB RTX 3090, Mosaic extends the supported context length by 32.34x over PyTorch and 33.4x over JetEngine (Sec. 5.2 in submission).
> *   **Achieving 2.27x–3.75x context extensions on tailored KV-cache stacks.** Applying KV caching introduces a lossy mechanism that compromises generation quality and structurally shifts the bottleneck to the persistent KV cache. While this naturally reduces Mosaic's optimization scope compared to lossless baselines, our approach remains highly complementary. Porting our techniques to Fast-dLLM, dInfer (both cited in our submission), and Elastic-Cache [2] extends supported context lengths by 3.75x, 3.00x, and 2.27x, respectively, under identical hardware constraints with zero latency degradation.
>
> We will explicitly clarify these consistent system-level advantages across both paradigms in the revised manuscript.
>
> [1] JetEngine. GitHub repository: https://github.com/Labman42/JetEngine.
>
> [2] Nguyen-Tri et al. "Attention is all you need for kv cache in diffusion llms." arXiv:2510.14973, 2025.
>
> **3. Batched Workloads and Real-World Serving (Weakness 2.2 & Key Question 2)**
>
> We are sorry for the confusion regarding real-world serving. While our initial evaluation focused on a batch size of 1 to push the absolute context length boundary, Mosaic’s scalability extends naturally to high-concurrency environments:
>
> *   **Theoretical Consistency:** Mosaic maximizes the total tokens processed under a hardware memory ceiling. Since the activation footprint scales linearly with total tokens (batch size $\times$ sequence length), extending context length is mathematically equivalent to extending batch size.
> *   **Experimental Validation:** New experiments confirm that Mosaic scales the maximum supported batch size by the same multipliers observed in context extension (averaging 32.34x, 25.41x, and 15.30x vs. PyNative, Mosaic-Torch, and Mosaic-Compile). Within the baselines' executable range, Mosaic exhibits no latency degradation.
>
> We will include these theoretical explanations and batched serving results in the revision.
>
> **4. Kernel Generalization (Key Question 3)**
>
> We are sorry for the confusion. The mask-only logits kernel is architecture-agnostic because it targets the final projection layer, which is structurally independent of the model backbone (e.g., MoE or dense). Our graph registrar automates this by dynamically mapping standard projection operators to our custom gather-GEMM kernel. We will clarify this automated mechanism in the revision.
>
> **5. Writing Improvement (Weakness 3)**
>
> We are sorry for the confusion and appreciate your feedback. Per your suggestions, we will comprehensively polish the writing and consolidate fragmented paragraphs to ensure a more rigorous and coherent logical flow throughout the manuscript.

---

> > ### Author Rebuttal · Reviewer_EE58 · 2026-04-04
> >
> > Thank the authors for the detailed response. I appreciate the newly added experiments. I believe the paper can be much stronger in terms of soundness with the authors' planned revisions.

---

> > > ### Author Response · Authors · 2026-04-07
> > >
> > > We are grateful for your constructive feedback and for acknowledging that our newly added experiments and detailed explanations addressed your concerns. Following your feedback, the comprehensive comparisons against existing dLLM-specific frameworks, the batched workload evaluations, and the clarifications on our architectural co-design and automated kernel mapping help highlight the practicality of our proposed system and further demonstrate how integrating these techniques effectively resolves the unique memory bottlenecks of dLLMs. These newly added experiments and clarifications significantly improve the paper, and we will carefully incorporate them into the revised version, alongside a comprehensive polish of the manuscript to ensure a rigorous logical flow.
> > >
> > > Thank you again for your thorough and constructive review.

---

### Official Review · Reviewer_FnFJ · 2026-03-12

**Soundness:** 2
**Presentation:** 2
**Significance:** 2
**Originality:** 1
**Overall Recommendation:** 3
**Confidence:** 4

**Summary:**

This paper proposes Mosaic, an inference technique for diffusion LLMs the reduce the memory footprint of activations during inference time. The method reduces the ratio of peak-to-average memory by 2.7x and increases the maximum supported sequence length by 15-32x.

**Compliance With Llm Reviewing Policy:**

Affirmed.

**Final Justification:**

The rebuttal addressed some of my concerns, and I have raised my score.

**Key Questions For Authors:**

1. How does Mosaic compare to previous dLLM optimization frameworks such as Fast-dLLM and the paper “Attention Is All You Need for KV Cache in Diffusion LLMs” ? Both in terms of methodological differences and actual measurement of performance.
2. How does Mosaic compare to simple one-line changes, e.g. in sharing activations?

**Limitations:**

yes

**Strengths And Weaknesses:**

Strengths:
* The paper correctly identifies that naive dLLMs do not have the same KV cache mechanism as autoregressive LLMs.
* It is nice to see systems work begin to emerge that can address the inference-time requirements of running diffusion LLMs.

Weaknesses:
* A major weakness is novelty and lack of comparison to existing work. For example, there is in fact already work applying KV cache to diffusion LLMS:
    * https://arxiv.org/abs/2510.14973 “Attention Is All You Need for KV Cache in Diffusion LLMs”
    * https://nvlabs.github.io/Fast-dLLM/ This paper is briefly cited in related work, but not compared to in the experiments
* Some claims are unsupported, such as "While general-purpose reuse mechanisms address activation memory to some extent (Pisarchyk & Lee, 2020; Ansel et al., 2024), they lack the global visibility to adapt to dLLMs’ dynamic peaks, which toggle between logits and feed-forward networks (FFNs).”
    * This is a key claim about why the paper needs to exist and why the dynamic memory management system needs to exist - but it is unclear how “dynamic peaks” are different than the standard activation overhead that occur during any inference forward pass.
    * In fact, it is a pretty common optimization to simply use a shared “activation tensor” and move results into it (something like `torch.mm(a, b, out=shared_tensor)`). It is unclear to the reviewer how much of the system reduces to replacing lines of the model with this.
* The observations in section 3 are not well grounded - for example, it is unclear to the reviewer what the difference is between observations 1 and 3.

The writing of the paper could use improvement:
* The abstract is both too long and does not contain enough detail about the method
    * There is too much discussion of dLLM basics in the first two paragraphs - those two paragraphs could be two sentences
    * Paragraph 3 of the abstract contains the key contributions, but the language is weak and jargon
* Generally, abstracts should not be more than one paragraph
* The introduction has the same problem - most of the contribution is in paragraph 4, but it is mostly jargon. It should also be a red flag to the authors that the abstract and introduction and nearly the same length, with nearly the same content

---

> ### Author Rebuttal · Authors · 2026-03-31
>
> We thank the reviewer for the constructive feedback. Mosaic addresses the severe memory bottlenecks standard inference systems face when scaling context lengths. Below, we clarify our advantages over recent KV-caching methods, analyze diffusion LLM (dLLM) memory peaks, and detail the architectural necessity of our global memory manager compared to simple in-place operations.
>
> **1. Discussion on KV-Cache Methods (Weakness 1 & Key Question 1)**
>
> We appreciate the suggestion to compare Mosaic against specialized dLLM frameworks. New experiments show Mosaic provides 2.27x to 3.75x orthogonal context extensions on KV-cache stacks, with zero latency degradation.
>
> Mosaic is fundamentally designed for standard, lossless dLLMs using full bidirectional attention, where memory bottlenecks strictly lie in intermediate activations. Conversely, applying Key-Value (KV) caching introduces a lossy mechanism that compromises generation quality (as noted in Fast-dLLM) and shifts bottlenecks toward persistent KV states. While this naturally reduces Mosaic's optimization scope compared to lossless baselines, our methods remain highly effective.
>
> Evaluated on a 24GB RTX 3090 with LLaDA-8B, we ported Mosaic onto Fast-dLLM, dInfer (both cited in our submission), and Elastic-Cache [1]. As shown below, Mosaic significantly extends supported context lengths across all frameworks:
>
> | Tailored KV-Cache Framework | Caching & Inference Paradigm           | Context Length Extension |
> | :-------------------------- | :------------------------------------- | :----------------------- |
> | Fast-dLLM                   | Prefix-cache and dynamic decoding      | 3.75x                    |
> | dInfer                      | Block-wise cache and dynamic inference | 3.00x                    |
> | Elastic-Cache               | Sliding-window active computation      | 2.27x                    |
>
> This confirms that by mitigating intermediate activation footprints, Mosaic serves as a robust and highly complementary architecture.
>
> [1] Nguyen-Tri et al. "Attention is all you need for kv cache in diffusion llms." arXiv:2510.14973, 2025.
>
> **2. Clarification on Dynamic Peaks and Shared Tensors (Weakness 2 & Key Question 2)**
>
> We are sorry for the confusion regarding our system implementation, and we appreciate the opportunity to distinguish our contributions from simple in-place operations (e.g., `torch.mm(a, b, out=shared_tensor)`).
>
> While in-place operations are necessary engineering mechanisms, they only offer localized memory reuse. They cannot resolve the shifting memory bottlenecks inherent to dLLMs, leading to severe external fragmentation and out-of-memory (OOM) errors. As shown in Sec. 5.3, a baseline using only shared tensors (the "+ global manager" baseline) performs worse than Mosaic. Mosaic extends far beyond one-line replacements via a co-designed runtime framework:
>
> *   **Identification of Dynamic Peaks:** We identified that dLLM activation memory peaks are not static; they dynamically toggle between the feed-forward network and the logits layer based on the varying mask ratio.
> *   **Global Virtual Memory Manager (VMM):** Our VMM manages the entire execution graph globally to fundamentally eliminate external memory fragmentation.
> *   **Online Lazy Chunking Optimizer:** This optimizer adaptively chunks computations to systematically tame the varying memory peaks that standard shared tensors cannot handle.
> *   **Custom Compute Operators:** We integrated dLLM-tailored operators (e.g., the mask-only logits kernel) to reduce the memory footprint.
>
> We will clarify the architectural necessity of our global memory management system over standard in-place operations in the revision.
>
>
> **3. Clarification on Observations 1 and 3 (Weakness 3)**
>
> We are sorry for the confusion regarding Observations 1 and 3. They describe dLLM memory characteristics from two distinct but related perspectives:
>
> *   **Observation 1 (Macro-Level Architectural Shift):** Identifies the system-level bottleneck. Due to step-wise bidirectional recomputation, transient intermediate activations, rather than the persistent KV-cache, form the primary memory bottleneck.
> *   **Observation 3 (Micro-Level Runtime Phenomenon):** Characterizes the dynamic runtime behavior *within* these activations. The peak memory footprint is not constant; it dynamically toggles between the feed-forward network and the logits layer based on the changing mask ratio during generation.
>
> We will explicitly clarify this hierarchical relationship in the revision.
>
> **4. Presentation Improvements (Weakness 4)**
>
> We are sorry for the confusion regarding the manuscript's presentation and sincerely appreciate your constructive feedback. Following your suggestions, we will comprehensively polish the writing, particularly the abstract and introduction, to ensure the text is concise and the core contributions are clearly communicated. We deeply value your review and will incorporate these improvements in our revision.

---

> > ### Author Rebuttal · Reviewer_FnFJ · 2026-04-03
> >
> > Thank you for the response, it has addressed my concerns.

---

> > > ### Author Response · Authors · 2026-04-07
> > >
> > > We are grateful for your constructive feedback and for acknowledging that our additional experiments and explanations addressed your concerns. Following your feedback, the new experimental comparisons with existing kv cache frameworks, along with the detailed clarifications regarding our system design and observations, help highlight the advantages of our system and clarify its behavior. These newly added experiments and clarifications significantly improve the paper, and we will carefully incorporate them into the revised version, alongside the overall writing improvements to ensure conciseness.
> > >
> > > Thank you again for your thorough and constructive review.

---

### Official Review · Reviewer_nYKm · 2026-03-13

**Soundness:** 3
**Presentation:** 3
**Significance:** 3
**Originality:** 3
**Overall Recommendation:** 5
**Confidence:** 2

**Summary:**

This paper studies the memory aspect of the inference of diffusion-based large language models (dLLMs) for long-context generation. This paper identifies that the highly optimized KV-Cache system for dLLMs doesn't work as effectively as the autoregressive models the due to its transient activation step, and the dynamic memory peaks of the dLLM workload.

Thus, this paper proposes MOSAIC as an inference system that captures the full computation cycle for memory-efficient long context dLLM inference. MOSAIC achieves ~2.71x reduction in memory peak-to-average ratio and extends the maximum supportable sequence length by at least 15.3 x, thus reducing the end-to-end latency by 2.5% - 55.4%.

**Compliance With Llm Reviewing Policy:**

Affirmed.

**Final Justification:**

The author addressed all my questions and concerns for the rebuttal. Specifically, they add 2 experiments to the multi-GPU setting.

**Key Questions For Authors:**

Could MOSAIC be generalized to multi-GPU or distributed inference settings?

How would the MOSAIC's improvement vary for GPUs with larger memory?

**Limitations:**

There is no limitations in the conclusion section, but there is an impact statement.

**Strengths And Weaknesses:**

Strength:
- The observations/insights of long-context workloads are quite informative.
- MOSAIC obtains 30x maximum sequence context length; this result is quite impressive.
- The presentation of this paper is clean and informative.

Weakness:
- It would be great to see if this system could be generalized to multi-GPUs.

---

> ### Author Rebuttal · Authors · 2026-03-31
>
> We thank the reviewer for the constructive feedback. Currently, dLLM research is divided into algorithmic quality and system performance. Supporting long-context capabilities is now a critical requirement, especially for AI Agents that demand extensive memory for complex reasoning. Driven by this necessity, we developed Mosaic to address the unique memory challenges that standard inference systems cannot manage.
>
> We appreciate the opportunity to further demonstrate Mosaic's scalability and robustness. Inspired by your comments, we have conducted new evaluations on multi-GPU and distributed inference settings. We also detail how our performance gains scale to GPUs with larger memory capacities. Our detailed responses are provided below.
>
> **1. Generalization to Multi-GPU and Distributed Inference Settings (Weakness 1 & Key Question 1)**
> We sincerely thank the reviewer for pointing out this critical aspect. Due to the strict time constraints of the rebuttal period, we dedicated our best effort to implement both Tensor Parallelism (TP) and Pipeline Parallelism (PP) prototypes within the Mosaic framework. As summarized in the table below, evaluated on the LLaDA model across 2 GPUs, Mosaic successfully achieves over 10x context length extension compared to the PyNative multi-GPU baselines:
>
> | Parallelism Strategy       | Hardware Setup | Context Length Extension (Mosaic vs. PyNative) |
> | :------------------------- | :------------- | :--------------------------------------------- |
> | Tensor Parallelism (TP2)   | 2 GPUs         | 12.30x                                         |
> | Pipeline Parallelism (PP2) | 2 GPUs         | 14.92x                                         |
>
> In distributed inference settings, the overall expansion ratio is naturally influenced by the necessary communication buffers. As the context scales to extreme lengths, the memory footprint of these fixed communication buffers (e.g., for inter-GPU tensor passing) grows linearly. While Mosaic effectively eliminates the computation activation bottleneck, these unoptimizable communication buffers eventually consume a significant proportion of the GPU memory, inherently bounding the theoretical upper limit of the current context extension. Moreover, our analysis suggests additional opportunities for optimization tailored to multi-GPU scenarios, such as implementing buffer chunking and distributed memory planning, which may yield an additional 10%--20% context length extension. Due to the rebuttal time limit, we leave these potential optimizations for future work.
>
> We deeply appreciate your constructive question. We will incorporate the analysis for these distributed scenarios and the new multi-GPU experimental results into the revised manuscript.
>
>
>
> **2. Impact of Larger GPU Memory on Performance Improvements (Key Question 2)**
> We expect Mosaic and larger-memory GPUs to be complementary. In Section 5.1 in our submission (Page 7), we have empirically observed this behavior across two distinct memory tiers: an RTX 3090 with 24GB VRAM and an A100 with 40GB VRAM.
>
> By analyzing the performance across these two hardware configurations, we find that the scaling of Mosaic's benefits with respect to GPU memory size is consistent. Because the memory footprint of intermediate activations scales linearly with the context length, intermediate activations remain the primary memory bottleneck regardless of the underlying memory capacity, allowing the relative expansion multiplier achieved by Mosaic to remain relatively stable. As reported in our Section 5.1, Mosaic  achieves an average context extension of 31.53x on the 24GB RTX 3090 and 33.14x on the 40GB A100 over the PyNative baseline. This consistent performance indicates that as GPU memory increases, our relative gains remain stable, demonstrating that our technique is robust to variations in hardware memory capacity.

---

> > ### Author Rebuttal · Reviewer_nYKm · 2026-04-03
> >
> > The author fully addressed my questions with additional experiments.

---

> > > ### Author Response · Authors · 2026-04-07
> > >
> > > We are grateful for your encouraging feedback and for acknowledging that our additional experiments addressed your questions. Following your feedback, the new multi-GPU evaluations and the in-depth analysis on hardware memory scaling help demonstrate the scalability and practical utility of our approach. These newly added experiments and analyses significantly improve the paper, and we will carefully incorporate them into the revised version.
> > >
> > > Thank you again for your thorough and constructive review.

---

### Official Review · Reviewer_1XLe · 2026-03-21

**Soundness:** 3
**Presentation:** 3
**Significance:** 2
**Originality:** 2
**Overall Recommendation:** 4
**Confidence:** 4

**Summary:**

MOSAIC is a systems solution for long-context diffusion LLM inference: (1) it replaces wasteful full-sequence logits with a mask-only fused kernel, (2) exposes whole-graph tensor lifetimes through an explicit graph registrar, (3) applies bottleneck-aware lazy chunking only when FFN or logits become the active memory peak, and (4) uses a virtual-memory-backed global allocator to avoid fragmentation.

**Compliance With Llm Reviewing Policy:**

Affirmed.

**Key Questions For Authors:**

1. Can you add either a direct comparison with an inference stack that supports dLLMs well like SGlang and JetEngine?

2. Do the large gains in supportable context length lead to measurable improvements on actual long-context tasks? Can the authors run some long-context benchmarks and check quality?

**Strengths And Weaknesses:**

1. The paper shows useful system insight: standard bidirectional dLLMs are activation-dominated in longcontext regime, with additional dynamic peaks that shift between FFN and logits as the mask ratio changes.

2. The method is training-free, lossless, and demonstrates very large context-capacity gains on commodity hardware. For long-context dLLM inference, that is a meaningful contribution.

Weakness:
1. Most comparisons are against internal or implementation-level baselines with compiler or kernel levels of optimizations (torch, torch.compile, vLLM), not initially designed for dLLMs. So the paper could position itself more clearly against broader systems that are tailored for like SGLang’s support for dLLMs, SDAR JetEngine.

2. The paper supports much longer context and lower per-step latency, backed by many system metrics. But could the authors provide more insights into those dLLMs’ longcontext capability? How do they work on long-context benchmarks like RULER? Where are quality metrics? Also how does it relate to inference block size?

3. The implementations add quite some system complexity. Unclear if really useful in practice.

---

> ### Author Rebuttal · Authors · 2026-03-31
>
> We thank the reviewer for their constructive feedback. As long-context capabilities become critical for dLLMs, standard inference systems struggle with severe memory bottlenecks. We developed Mosaic to address these specific challenges. Inspired by your suggestions regarding baselines, quality metrics, and practical utility, we conducted further evaluations and present the results below.
>
> **1. Comparisons with Tailored Inference Stacks (Weakness 1 & Key Question 1)**
>
> We categorized dLLM inference setups into two paradigms to provide a comprehensive evaluation against tailored systems:
>
> | Method Name                                    | Official PyTorch | JetEngine | Fast-dLLM | dInfer | Elastic-Cache |
> | :--------------------------------------------- | :--------------- | :-------- | :-------- | :----- | :------------ |
> | Context Length Extension (Mosaic vs. Baseline) | 32.34x           | 33.4x     | 3.75x     | 3.00x  | 2.27x         |
>
> *   **Achieving >32x context extensions on standard lossless baselines.** Standard implementations (official PyTorch, JetEngine [1]) lack KV caches, making intermediate activations their strict memory bottleneck (Sec. 3 in submission). By systematically eliminating unmanaged activation peaks and fragmentation via lazy chunking and global memory management, Mosaic prevents early Out-Of-Memory (OOM) errors. Evaluated on a 24GB RTX 3090, Mosaic extends the supported context length by 32.34x over PyTorch and 33.4x over JetEngine (Sec. 5.2 in submission).
> *   **Achieving 2.27x–3.75x context extensions on tailored KV-cache stacks.** Applying KV caching introduces a lossy mechanism that compromises generation quality and structurally shifts the bottleneck to the persistent KV cache. While this naturally reduces Mosaic's optimization scope compared to lossless baselines, our approach remains highly complementary. Porting our techniques to Fast-dLLM, dInfer (both cited in our submission), and Elastic-Cache [2] extends supported context lengths by 3.75x, 3.00x, and 2.27x, respectively, under identical hardware constraints with zero latency degradation.
>
> We will explicitly clarify these consistent system-level advantages across both paradigms in the revised manuscript.
>
> [1] JetEngine. GitHub repository: https://github.com/Labman42/JetEngine.
>
> [2] Nguyen-Tri et al. "Attention is all you need for kv cache in diffusion llms." arXiv:2510.14973, 2025.
>
> **2. Long-Context Quality Metrics and Block Size (Weakness 2 & Key Question 2)**
>
> We are sorry for the confusion regarding long-context performance. Mosaic is strictly an underlying system optimization that does not alter the attention mechanisms, weights, or internal precision of diffusion LLMs (dLLMs), such as LLaDA and Dream. Consequently, our generated results and quality metrics on benchmarks like RULER are perfectly identical to the base models.
> Our core contribution is alleviating memory and latency bottlenecks to process significantly longer contexts under the same hardware constraints. Enhancing a model's intrinsic reasoning capability, or tuning the inference block size to adjust generation quality, are algorithmic endeavors. These are orthogonal to Mosaic, which simply serves as the infrastructure to efficiently execute these models regardless of the chosen block size.
> We will clarify our system-level scope, precision preservation, and the role of block size in the revision.
>
> **3. System Complexity vs. Practical Utility (Weakness 3)**
>
> We are sorry for the confusion regarding system complexity. While implementation requires non-trivial engineering, this complexity is deeply encapsulated and entirely abstracted from the end-user. Mosaic's practical utility is concretely demonstrated as follows:
>
> *   **Minimal Runtime Overhead:** Our dynamic memory management introduces negligible latency (Sec. 5.3). Lightweight search and first-fit planning reduce management overhead to mere fractions of traditional methods (search is 0.28%–0.32% of brute-force; planning is 0.1%–4.3% of ILP). Moreover, efficiency gains from optimizations like the mask-only logits kernel far outweigh this minuscule cost. Consequently, Mosaic actually *reduces* overall per-step latency across all baselines—by 5.9%–55.4% vs. PyNative, and 2.5%–24.9% vs. Mosaic-Torch and Mosaic-Compile (Sec. 5.2). This proves that system complexity does not degrade practical performance.
> *   **Usability and Open Source:** The framework easily accommodates various hardware backends. Developers only need 50–100 lines of code to register a new model via our graph register (Sec. 4.3), seamlessly integrating it into standard workflows while completely hiding all internal memory management complexities. To further contribute to the community, we will fully open-source the Mosaic codebase in the near future.

---

> > ### Author Rebuttal · Reviewer_1XLe · 2026-04-07
> >
> > good rebuttal, thanks!

---

> > > ### Author Response · Authors · 2026-04-08
> > >
> > > We are grateful for your constructive feedback and for acknowledging that our additional experiments and explanations addressed your concerns. Following your feedback, the direct comparisons with tailored inference stacks, alongside the detailed explanations regarding preserving exact model outputs and framework usability, help highlight the practical utility and robustness of our proposed system. These newly added experiments and clarifications significantly improve the paper, and we will carefully incorporate them into the revised version.
> > >
> > > Thank you again for your thorough and constructive review.

---

### Decision · Program_Chairs · 2026-04-30

**Decision:**

Accept (regular)

**Comment:**

The paper studies the problem of serving dLLMs with long context. Authors propose a system of design choices to boost context window by 30x, including custom fused gather-GEMM, lazy chunking optimizer, global memory manager. Some reviewers characterized work as standard ideas adopted to dLLMs.

Reviewers rated soundness good, presentation good-to-fair, significance fair-to-good, and originality fair-to-good, reaching mixed consensus (one Accept, two Weak Accept, one initial Weak Reject that improved after rebuttal).

As main strength, reviewers mentioned the impressive practical gains of 15-32x longer context lengths on commodity hardware while remaining training-free and lossless, along with insightful characterization of dLLM memory bottlenecks.

As key weakness, reviewers questioned the incremental nature of the techniques (standard systems optimizations applied to dLLMs) and limited initial comparisons to specialized dLLM frameworks.

All reviewers fully acknowledged the rebuttal and resolved concerns (scores raised or maintained).

The paper is technically sound and well-engineered (a bit less researchy) with valuable systems contributions for emerging dLLM inference, making it useful to the ICML community. Given this, my recommendation is Weak Accept.